# NF-κB Transcriptional Activity Indispensably Mediates Hypoxia–Reoxygenation Stress-Induced microRNA-210 Expression

**DOI:** 10.3390/ijms24076618

**Published:** 2023-04-01

**Authors:** Gurdeep Marwarha, Katrine Hordnes Slagsvold, Morten Andre Høydal

**Affiliations:** 1Group of Molecular and Cellular Cardiology, Department of Circulation and Medical Imaging, Faculty of Medicine and Health, Norwegian University of Science and Technology (NTNU), 7034 Trondheim, Norway; 2Department of Cardiothoracic Surgery, St. Olavs University Hospital, 7030 Trondheim, Norway

**Keywords:** NF-κB, miR-210, hypoxia–reoxygenation, AC-16 cardiomyocytes, histone modification, RNA polymerase II

## Abstract

Ischemia–reperfusion (I-R) injury is a cardinal pathophysiological hallmark of ischemic heart disease (IHD). Despite significant advances in the understanding of what causes I-R injury and hypoxia–reoxygenation (H-R) stress, viable molecular strategies that could be targeted for the treatment of the deleterious biochemical pathways activated during I-R remain elusive. The master hypoxamiR, microRNA-210 (miR-210), is a major determinant of protective cellular adaptation to hypoxia stress but exacerbates apoptotic cell death during cellular reoxygenation. While the hypoxia-induced transcriptional up-regulation of miR-210 is well delineated, the cellular mechanisms and molecular entities that regulate the transcriptional induction of miR-210 during the cellular reoxygenation phase have not been elucidated yet. Herein, in immortalized AC-16 cardiomyocytes, we delineated the indispensable role of the ubiquitously expressed transcription factor, NF-κB (nuclear factor kappa-light-chain-enhancer of activated B cells) in H-R-induced miR-210 expression during cellular reoxygenation. Using dominant negative and dominant active expression vectors encoding kinases to competitively inhibit NF-κB activation, we elucidated NF-κB activation as a significant mediator of H-R-induced miR-210 expression. Ensuing molecular assays revealed a direct NF-κB-mediated transcriptional up-regulation of miR-210 expression in response to the H-R challenge that is characterized by the NF-κB-mediated reorchestration of the entire repertoire of histone modification changes that are a signatory of a permissive actively transcribed miR-210 promoter. Our study confers a novel insight identifying NF-κB as a potential novel molecular target to combat H-R-elicited miR-210 expression that fosters augmented cardiomyocyte cell death.

## 1. Introduction

Myocardial ischemia-reperfusion (I-R) injury is a cardinal pathophysiological hallmark of ischemic heart disease (IHD), the leading cause of global morbidity and mortality. At the cellular level, either acute or chronic myocardial ischemia-elicited hypoxia stress induces cardiomyocyte cell death that culminates into irremediable myocardial damage that may clinically manifest as acute myocardial infarction (AMI). Timely coronary reperfusion is the best clinical recourse to curtail the irrevocable deleterious impact on the myocardium but paradoxically accentuates the prevailing myocardial injury further, resulting in augmented cardiomyocyte death, a pathophysiological hallmark that characterizes myocardial I-R injury. The master *hypoxamiR*, microRNA-210 (miR-210), is unanimously accorded as a major molecular determinant and an indispensable cellular entity in cellular adaptation to hypoxia stress [1,2,3,4,5,6,7]. Cellular response to myocardial I-R injury is biphasic, with an adaptive response during the ischemia/cellular hypoxia phase that is followed by a deleterious molecular reprogramming of the intracellular milieu that confers greater vulnerability of the myocardial tissue to cellular injury and cardiomyocyte death upon reperfusion/cellular reoxygenation in the quest to resuscitate the infarcted myocardium. The transcriptional regulation of miR-210, the master *hypoxamiR*, under hypoxia conditions is well delineated, with HIF1α (hypoxia-inducible factor 1-alpha) indispensably mediating the induction of miR-210 expression during hypoxia [3,4,8,9,10,11,12,13,14,15]. However, there is a significant void in the understanding of molecular entities and the cellular signaling pathways that govern and regulate miR-210 expression during the myocardial reperfusion/cellular reoxygenation phase. Our recent study in human AC-16 cardiomyocytes has demonstrated that the cellular hypoxia-induced miR-210 expression is sustained, and even further enhanced, during the cellular reoxygenation phase [16] when HIF1α expression and transcriptional activity is completely abrogated [17,18,19,20]. Furthermore, this hypoxia–reoxygenation (H-R)-induced miR-210 expression is deleterious and indispensably contributes to the H-R-induced apoptotic cell death of the AC-16 cardiomyocytes [16]. In light of this unprecedented finding, it behooves to characterize the upstream transcriptional regulatory events and transcription factors that play a seminal role in driving miR-210 expression during the cellular reoxygenation phase. The in-silico analysis of the miR-210 gene promoter has unveiled conserved binding sites for the transcription factor NF-κB (Nuclear Factor kappa-light-chain-enhancer of activated B cells) [21], a necessary molecular component of the orchestrated cellular response to the H-R challenge in cells as well as pathophysiological responses to myocardial I-R injury [22,23]. In this study, we characterized and dissected the functional role of NF-κB activation in the regulation of H-R-induced miR-210 expression and cogently demonstrated the indispensable contribution and mediation of direct NF-κB transcriptional activity in the H-R-induced enhanced miR-210 expression. We further elucidated and exhaustively delineated the direct NF-κB-mediated epigenetic regulation of the miR-210 gene promoter, the NF-κB-associated transcriptional regulatory mechanisms, as well as the ensuing downstream molecular events that indispensably mediate the H-R-induced and NF-κB-mediated transcriptional upregulation of miR-210 expression.

## 2. Results

### 2.1. Hypoxia–Reoxygenation (H-R) Insult Increases the Binding of NF-κB in the miR-210 Proximal Promoter

The miR-210 gene (Gene ID: 406992, https://www.ncbi.nlm.nih.gov/gene/406992) (date of access—20 November 2019) is transcribed from the genomic locus GRCh38.p14 on chromosome 11p 15.5 (NC_00 0011.10:568089-568198 Homo sapiens chromosome 11, GRCh38.p14) [6,24] (https://www.ncbi.nlm.nih.gov/nuccore/NC_000011.10?from=568089&to=568198&report=genbank&strand=true, 20 November 2019). The genomic size of pri-miR-210 is predicted to be 2927 base-pairs (bp) with the functional proximal promoter elements residing within 500 bp of the transcription start site (TSS) [6,25]. To characterize and dissect the functional role of NF-κB in H-R-induced miR-210 expression, we first performed an in silico analysis of the miR-210 promoter, using ALGGEN PROMO [26,27] and the TRANSFAC database, that unveiled multiple conserved transcription factor binding sites for p65 NF-κB (p65 subunit of NF-κB, also called Rel A) (Appendix A). Next, we validated the NF-κB activation as a molecular consequence of H-R. To this end, we determined the translocation of the p65 (Rel A) and p50 (Rel B) subunits of NF-κB from the cytosol into the nucleus and the direct NF-κB transcriptional activity in AC-16 cells subjected to H-R. We segregated the cytosolic and nuclear compartments using cellular fractionation (Section 4.3). The integrity of the cytosolic fraction was validated by the presence of HSP90-β (heat shock 90kD protein 1, Beta) concomitant with the absence of histone H3, while the corollary criteria, i.e., the absence of HSP90-β concomitant with the presence of histone H3, was used to validate the integrity of the nuclear fraction (Appendix A). Quantitative sandwich ELISA immunoassays revealed that H-R elicited a significant increase in the abundance of p65 NF-κB and p50 NF-κB in the nuclear compartment concomitant with the relative decrease in abundance of the respective subunits in the cytosolic compartment (Appendix A). The NF-κB transcriptional activity reporter assay also unveiled that H-R evoked a commensurate increase in NF-κB transcriptional activity as a consequence of enhanced p65 NF-κB and p50 NF-κB translocation into the nucleus (Appendix A). Having established H-R challenge-induced NF-κB activation, we delved into empirically characterizing the role of NF-κB activation in H-R-induced miR-210 expression. To this end, we first performed an miR-210 promoter pull-down assay using a reverse chromatin immunoprecipitation (R-ChIP) approach [28,29,30], followed by Western blot-based immunodetection of the miR-210 proximal promoter-bound p65 NF-κB and p50 NF-κB. Quantitative densitometry-coupled Western blot analysis showed that the H-R challenge elicited a profound enrichment of the p65 NF-κB (Figure 1A,B) and p50 NF-κB (Figure 1A,C) in the reverse crosslinked miR-210 promoter pull-down fragment. To further characterize the functional role of NF-κB activation in the H-R-induced transcriptional induction of miR-210 expression, we adopted the corollary approach and performed chromatin immunoprecipitation (ChIP) with tandem ELOHA (enzyme-linked oligonucleotide hybridization assay) [31] analysis to determine the abundance of the miR-210 *proximal* promoter fragment in the p65 NF-κB-immunoprecipitated chromatin. Tandem ChIP-ELOHA analysis revealed a pronounced increase in p65 NF-κB binding to the κB response element in the miR-210 *proximal* promoter in response to the H-R challenge (Figure 1D), relative to the non-existent p65 NF-κB binding under basal normoxia conditions (Figure 1D). The relative abundance of β-actin and the chromatin-associated TBP (TATA-box binding protein) in the native cellular lysates was determined using Western blotting (Figure 1A) to validate the equitable input experimental samples subjected to the miR-210 promoter pull-down. To further focus on the role of NF-κB transcriptional activity as a significant mediator of the H-R-induced increase in miR-210 expression, we determined the H-R-driven miR-210 expression in an experimental paradigm characterized by the *competitive* inhibition of the NF-κB signaling pathway. To this end, we inhibited the NF-κB pathway by ectopically expressing, either the *dominant active* mutant of IκBα (*da*-IκBα) (characterized by the IκBα-S32A/S36A mutant) or the *dominant negative* mutant of IKKα/β (*dn*-IKKα/β) (characterized by the kinase-dead IKKα-K44M and IKKβ-K44A mutants). Enzyme-coupled miR-210 hybridization immunoassays revealed that the H-R-induced increase in miR-210 expression was significantly mitigated in AC-16 cells ectopically expressing either *da*-IκBα (Figure 2A) or *dn*-IKKα/β (Figure 2B). The mitigating effects of *da*-IκBα or *dn*-IKKα/β in the H-R-induced increase in miR-210 expression were reflected by a commensurate decrease in p65-NF-κB binding to the κB response element in the miR-210 proximal promoter. Ectopic expression of *da*-IκBα (Figure 2C) or *dn*-IKKα/β (Figure 2D) completely abrogated the H-R-induced increase in p65-NF-κB binding to the κB response element in the miR-210 *proximal* promoter (Figure 2C,D).

We further analyzed the reverse crosslinked miR-210 promoter pull-down lysates to determine the abundance of p65 NF-κB-associated transcriptional coactivators and transcriptional corepressors being enriched at the κB response element in the miR-210 *proximal* promoter. To this end, we determined the abundance of miR-210 promoter-bound NF-κB transcriptional coactivators, p300 and CBP (CREB-binding protein), concomitant with miR-210 promoter-bound NF-κB transcriptional corepressors, NCoR1 (nuclear receptor co-repressor 1) and SMRT (silencing mediator of retinoic acid and thyroid hormone receptor). Quantitative sandwich ELISA immunoassays revealed that the H-R challenge elicited a significant enrichment of p300 (Figure 3A) and CBP (Figure 3B) concomitant with a decrease in the abundance of NCoR1 (Figure 3C) and SMRT (Figure 3D) at the miR-210 *proximal* promoter.

Furthermore, this H-R challenge elicited enrichment of p300 (Figure 3A) and CBP (Figure 3B) with a concomitant decrease in the abundance of NCoR1 (Figure 3C) at the miR-210 *proximal* promoter and was contingent on NF-κB activation, as the ectopic expression of *da*-IκBα significantly ablated these effects. Collectively, these data demonstrate that the H-R challenge induces miR-210 expression via an increase in the binding of NF-κB to the miR-210 promoter and the subsequent recruitment of the NF-κB transcription-associated chromatin modifiers to the miR-210 proximal promoter. 

### 2.2. H-R Induced NF-κB Transcriptional Activation Increases the Recruitment and Occupancy of Active RNA Polymerase II (RNAPII) at the miR-210 Promoter

The NF-κB-mediated transactivation of the promoters of target Class II genes engages the recruitment of the basal transcriptional machinery composed of the RNA polymerase II complex (RNAPII) in association with other transcription factors and transcriptional coactivators that constitute the pre-initiation complex (PIC). We determined the RNAPII occupancy at the transcription start site (TSS) in the miR-210 proximal promoter. To this end, we performed tandem ChIP-ELOHA analysis determining the abundance of the largest subunit and the catalytic component of RNAPII as RPB1 (RNA-directed RNA polymerase II subunit RPB1) [32,33] recruited at the miR-210 proximal promoter that serves as a surrogate for the RNAPII occupancy at the TSS of miR-210. Our ChIP-ELOHA data showed that H-R induces a significant enrichment of RPB1 at the miR-210 proximal promoter (Figure 4A). However, this H-R-induced profound enrichment of RPB1 at the miR-210 proximal promoter was not altered in cells ectopically expressing the *da*-IκBα mutant (Figure 4A), thereby suggesting that NF-κB activation does not regulate the H-R-induced increase in RNAPII occupancy at the miR-210 proximal promoter. The increase in RNAPII occupancy at the miR-210 proximal promoter cannot be unequivocally inferred as an increase in active transcription of miR-210 as the associated RNAPII complex could be in a *quiescent paused* state [34,35,36,37]. The C-terminal domain (CTD) of RPB1 contains multiple copies of consensus *heptad repeats* (Y_1_S_2_P_3_T_4_S_5_P_6_S_7_) [38] that are phosphorylated [39,40] in the RPB1 subunit recruited to the promoter of active genes [41,42,43]. Thus, the enrichment of the CTD-phosphorylated RPB1 (Ser^2^/Ser^5^) subunit of RNAPII, in the proximal promoter of target genes, is an accurate surrogate measure of RNAPII-mediated transcription elongation and an inverse index of RNAPII proximal promoter *pausing* [44]. Ergo, we further performed ChIP-ELOHA analysis to determine the enrichment of CTD-phosphorylated RPB1 recruited to the proximal promoter of miR-210. The ChIP-ELOHA data revealed a profound increase in the occupancy of Ser^2^/Ser^5^ CTD-phosphorylated RPB1 at the proximal promoter of miR-210 (Figure 4B). Furthermore, the H-R challenge-elicited increase in the occupancy of Ser^2^/Ser^5^ CTD-phosphorylated RPB1 in the miR-210 proximal promoter was contingent on NF-κB activation as the ectopic expression of the *da*-IκBα mutant significantly mitigated the H-R challenge response (Figure 4B).

To further corroborate that the miR-210 promoter-associated RNAPII complex was in a state of active transcription elongation in contrast to the *quiescent paused* state, we determined the abundance of the negative elongation factor (NELF) complex and the associated DSIF (5,6-dichloro-1-β-d-ribofuranosylbenzimidazole [DRB] sensitivity-inducing factor) complex recruited to the CTD of the RPB1 subunit of RNAPII associated with the miR-210 promoter. The NELF and DSIF protein complexes cooperatively regulate transcription elongation by binding to the CTD of the RPB1 subunit of RNAPII, resulting in the stalling or pausing of RNAPII elongation [45,46,47]. We, therefore, deemed the abundance of RBP1-bound NELF and DSIF protein complexes as a surrogate measure of the index of RNAPII in the *paused* state. We determined the relative abundance of the RPB1-bound NELF-A subunit of the tetrameric NELF complex [48] in reverse crosslinked miR-210 promoter pull-down immunoprecipitates using Western blot-coupled densitometric analysis (Figure 4C,D). The H-R challenge completely abolished the basal association of NELF-A with the CTD of the miR-210 promoter-bound RPB1 subunit of the RNAPII complex (Figure 4C,D). The H-R challenge evoked a loss of NELF-A association with the CTD of the miR-210 promoter-bound RPB1 subunit of the RNAPII complex, which was contingent on NF-κB activation as the ectopic expression of the *da*-IκBα mutant conferred significant refractoriness to this H-R-induced abolishment of the binding of NELF-A to the CTD of the miR-210 promoter-bound RPB1 subunit of the RNAPII complex (Figure 4C,D). We subsequently determined the association of the DSIF protein complex, an indispensable regulator of RNAPII-mediated transcription elongation [46]. The DSIF protein complex is a heterodimer of SPT4 (suppressor of Ty 4, also known as p14 DSIF) and SPT5 (suppressor of Ty 5, also known as p160 DSIF) subunits [49,50,51] that acts cooperatively with the NELF complex to repress RNAPII-mediated transcription elongation. We found no relative changes in the abundance of SPT4 and SPT5 in the reverse crosslinked miR-210 promoter pull-down lysates emanating from the entire gamut of the spectrum of the experimental groups (Figure 4C) (quantitative densitometry was not performed). The phosphorylation of the SPT5 subunit at multiple threonine residues within the CTR (C-terminal repeat) domain results in the dissociation of the NELF complex from the CTD of the RPB1 subunit of the paused RNAPII and the conformational change in the DSIF complex that culminates in the functional switching of the DSIF complex into an RNAPII-mediated transcription elongation activator [52]. We, therefore, deemed the abundance of p-Thr residues in the CTR domain of SPT5 as a bona fide accurate measure of the status of RNAPII-mediated transcription elongation. To this end, we immunoprecipitated out the entire pool of SPT5 in the reverse crosslinked miR-210 promoter pull-down lysates followed by the Western blotting-based determination of the extent of p-Thr residues in the immunoprecipitated SPT5 (Figure 4E,F). Western Blotting (Figure 4E) coupled with quantitative densitometric analysis (Figure 4F) of the immunoprecipitated SPT5 emanating from the reverse crosslinked miR-210 promoter *pull-down* lysates unveiled a pronounced increase in the p-Thr SPT5 expression levels in response to the H-R challenge (Figure 4E,F), a signatory of RNAPII-mediated transcription elongation. Furthermore, this H-R challenge-evoked increase in the abundance of p-Thr SPT5 was contingent on NF-κB activation as the ectopic expression of the *da*-IκBα mutant significantly mitigated this response (Figure 4E,F). Collectively, these data show that H-R challenge-elicited NF-κB activation significantly mediates H-R-induced miR-210 expression through the enhancement of the recruitment and occupancy of active RNAPII at the miR-210 proximal promoter. 

### 2.3. H-R-Induced NF-κB Transcriptional Activation Results in Histone Modification Changes That Are a Signatory of an Active miR-210 Promoter

Chromatin remodeling through post-translational histone modifications is an inherent molecular hallmark of transcriptional regulation [53,54,55,56]. The acetylation of designated lysine residues in histone H3 and histone H4, constituting the nucleosomes in target genes, is considered a molecular signature of *permissive* chromatin that characterizes actively transcribing gene promoters [53,54,55,56,57,58,59,60]. We characterized the magnitude of miR-210 promoter *permissiveness* by determining the acetylation status of the known N-terminal residues of histone H3 and histone H4 that make the gene promoters more amenable for promoter transactivation and the subsequent transcription of the gene [61,62]. The acetylation of histone H3 at Lys^9^ (H3K9ac), Lys^14^ (H3K14ac), Lys^18^ (H3K18ac), and Lys^27^ (H3K27ac) and histone H4 at Lys^5^ (H4K5ac), Lys^8^ (H4K8ac), and Lys^12^ (H4K12ac) is a molecular correlate of active *permissive* chromatin and associated with active gene promoters [54,60]. We performed ChIP-ELOHA analysis to determine the enrichment of the aforementioned acetylated lysine residues in the N-terminal tails of histone H3 and histone H4. The ChIP-ELOHA analysis revealed that the H-R challenge elicits a significant increase in the acetylation of histone H3 at Lys^9^ (H3K9ac) (Figure 5A), Lys^14^ (H3K14ac) (Figure 5B), Lys^18^ (H3K18ac) (Figure 5C), and Lys^27^ (H3K27ac) (Figure 5D) concomitant with an augmentation in the acetylation of histone H4 at Lys^5^ (H4K5ac) (Figure 6A), Lys^8^ (H4K8ac) (Figure 6B), and Lys^12^ (H4K12ac) (Figure 6C) in the nucleosomes that envelope the miR-210 proximal promoter. This H-R-evoked increase in the acetylation of the signatory lysine residues in histone H3 and histone H4 that envelope the miR-210 proximal promoter is dependent on NF-κB activation as ectopic expression of the *da*-IκBα mutant significantly attenuates this effect (Figure 5A–D and Figure 6A–C).

Next, we determined the methylation status of the known histone H3 and histone H4 residues in the nucleosomes that envelope the miR-210 proximal promoter [54,60]. Histone methylation in critical lysine residues in the promoters of genes bears a unique signature conferring the promoter to be in either the *permissive* active state or the *refractory* repressed (inactive) state [54,60]. The trimethylation of histone H3 at the Lys^4^ residue (H3K4me3) and Lys^36^ residue (H3K36me3) is considered a molecular signature of a *permissive* active promoter that is associated with transcriptional activation. 

The ChIP-ELOHA analysis of the nucleosomes that envelope the miR-210 proximal promoter unveiled that the H-R challenge evokes significant enrichment in the levels of H3K4me4 (Figure 7A) and H3K36me3 (Figure 7B), a characteristic molecular hallmark of a *permissive* active promoter. Furthermore, NF-κB activation is a significant molecular mediator of this H-R-induced increase in abundance of H3K4me4 (Figure 7A) and H3K36me3 (Figure 7B) within the nucleosomes that envelope the miR-210 promoter as the ectopic expression of the *da*-IκBα mutant significantly mitigates this response. The trimethylation of histone H3 at the Lys^9^ residue (H3K9me3) and Lys^27^ residue (H3K27me3) is considered a molecular hallmark of a *refractory* repressed promoter associated with transcriptional repression. The ChIP-ELOHA analysis of the relative enrichment of the trimethylation of histone H3 at the Lys^9^ residue (H3K9me3) and Lys^27^ residue (H3K27me3) revealed that the H-R challenge elicits a significant depletion in the abundance of H3K9me3 (Figure 7C) and H3K27me3 (Figure 7D), a molecular hallmark of a *refractory* repressed promoter. The ectopic expression of the HA-tagged *da*-IκBα mutant in the respective experimental inputs subjected to the histone modification ChIP assays was validated using a sandwich ELISA immunoassay performed against the HA tag (Appendix A). NF-κB transcriptional activity (Appendix A) was determined in the experimental inputs to corroborate and validate the translative effects of the ectopic expression of the *da*-IκBα mutant. Taken together, these data demonstrate that NF-κB activation significantly mediates the H-R challenge-induced increase in chromatin remodeling and histone modification changes that are signatories of an open *permissive* chromatin that confers a transcriptionally active miR-210 promoter.

## 3. Discussion

The unprecedented identification of NF-κB as a significant modulator of miR-210 expression in response to cellular reoxygenation has unveiled a new dimension of miR-210 biology that bestows significant ramifications on our comprehension of the biochemical and molecular milieu shaping the cellular response to an H-R insult. The cellular mechanisms and molecular entities that mediate miR-210 expression during myocardial reperfusion/cellular reoxygenation are yet to be elucidated and delineated, thereby warranting a detailed investigation for a multitude of reasons. Firstly, reoxygenation-driven miR-210 expression exacerbates apoptotic cardiomyocyte cell death [16]; therefore, it behooves to unveil and dissect the upstream molecular mechanisms that drive the reoxygenation-induced miR-210 expression. Secondly, given that miR-210 reprograms the entire spectrum of the biochemical and molecular response that constitute the cellular (mal)adaptive response to an I-R/H-R insult, reoxygenation-driven miR-210 expression could be an indispensable component of secondary pathophysiological changes. Contemporary evidence to date has unveiled a dichotomous view pertaining to the role of miR-210 in I-R/H-R injury, characterized by studies ascribing a protective role of miR-210 in response to I-R/H-R injury [63,64,65,66,67], while other studies implicate miR-210 as a requisite molecular mediator of the pernicious I-R/H-R injury response [68,69]. This can be attributed to disparities in the experimental paradigm and the heterogeneity of the disease model systems under investigation in the context of I-R injury. Interestingly, recent work unveiled a bimodal effect of miR-210 on apoptotic cell death in AC16 cardiomyocytes, with miR-210 mitigating apoptotic cell death during hypoxia and exacerbating apoptotic cell death during the reoxygenation phase [16], where the cellular mechanism mediating the miR-210-induced apoptotic cell death under reoxygenation conditions was ascribed to the activation of the *extrinsic* apoptotic cascade [16]. Ergo, it is imperative to distinguish between the hypoxia/ischemia-elicited transcriptional induction of miR-210 versus the cellular reoxygenation/tissue reperfusion-evoked transcriptional induction of miR-210. While the hypoxia/ischemia-evoked HIF1α-mediated transcriptional induction of miR-210 has been exhaustively characterized and well delineated [17,18,19,20], the unprecedented findings from this study characterized the implicit role of NF-κB as an indispensable molecular entity and transcriptional component that orchestrates the cellular reoxygenation-elicited transcriptional induction of miR-210.

Cellular H-R and pathological I-R injury elicit a significant increase in proinflammatory cytokine expression [70,71,72,73] that invokes the actuation of the death receptor (DR) signaling pathway [74,75,76,77,78], culminating in *extrinsic* apoptosis cascade-mediated cardiomyocyte cell death. NF-κB is widely considered an indispensable signaling molecule in orchestrating the cellular inflammatory milieu and modulating the ensuing apoptotic cell death in response to H-R and I-R injury [22,23]. Interestingly, accumulating contemporary evidence strongly implicates increased miR-210 expression in inducing a proinflammatory phenotype in different cell types [79,80]. Thus, it does warrant empirically inquiring and delineating whether the NF-κB- and the miR-210-induced cellular inflammation and the modulation of the ensuing apoptotic cell death during the cellular reoxygenation phase are molecular constituents of the same biochemical rubric, that is set-in responses to cellular H-R insult. The findings emanating from this study partially resolve this intricate molecular propinquity between cellular reoxygenation-induced NF-κB activation and enhanced miR-210 expression, establishing NF-κB as an upstream regulator of cellular reoxygenation-induced augmented miR-210 expression. In light of our findings, it is important to acknowledge that recent evidence shows that miR-210 overexpression drives NF-κB transcriptional responses and, thereby, is upstream of NF-κB signaling activation, albeit in immortalized PC-3 prostate cancer cells [81]. It remains, however, to be empirically determined whether NF-κB activation is also downstream of the cellular reoxygenation-induced miR-210 expression, thereby forming a positive feedback loop whereby cellular reoxygenation-induced miR-210 expression reinforces and exponentially amplifies its expression through downstream NF-κB signaling activation.

In propounding the putative positive feedback loop between cellular reoxygenation-induced miR-210 expression and NF-κB signaling activation, it is imperative to expound the (patho)physiological role of NF-κB signaling activation as its manifested biochemical and molecular impact is not unequivocally delineated during cellular reoxygenation. Multiple lines of research evidence have implicated NF-κB signaling, both as a primary molecular instigator of H-R damage/I/R injury as well as an indispensable component of the protective adaptive cellular response to H-R damage/I-R injury. Thus, the unveiling of the deleterious miR-210 expression during H-R injury as a direct NF-κB target serves as a pedestal for ensuing studies to determine the role of NF-κB-driven miR-210 expression in the NF-κB-elicited modulation of apoptotic cell death in response to an H-R challenge. Furthermore, considering our novel findings that unveil miR-210 as a bona fide direct NF-κB target gene during cellular reoxygenation, it is imperative to chart the upstream cellular and molecular inputs that induce NF-κB activation in response to cellular reoxygenation. While the extracellular proinflammatory cytokine-instigated death receptor (DR)-mediated signaling pathway could represent the primary mode of NF-κB activation in response to an H-R challenge, other coincident intracellular stimuli and kinase signaling pathways may directly modulate NF-κB activation. The serine/threonine kinase glycogen synthase kinase 3 beta (GSK3β) is considered the focal node of the *convergence* and *divergence* of multiple signaling cascades that regulate and effectuate (mal)adaptive responses to a multitude of noxious stimuli, including H-R challenges [82,83]. A significant volume of evidence from contemporary studies has highlighted concerted and dynamic crosstalk between GSK3β kinase activity and NF-κB activation [84,85,86]. Further studies are warranted to address this specific gap in knowledge by delving into the effects of the H-R-induced modulation of GSK3β kinase activity and the ensuing translative effects on NF-κB-driven miR-210 expression.

## 4. Materials and Methods

### 4.1. Cell Culture and Treatments

Human AC-16 cardiomyocyte cells (EMD Millipore/Merck Millipore/Merck Life Sciences, Catalogue # SCC109, Darmstadt, Germany, RRID:CVCL_4U18) were cultured and sub-cultured in the standard maintenance medium—Dulbecco’s modified Eagle medium (DMEM): Ham’s F12 (1:1; *v*/*v*) with 2 mM glutamine, 12.5% fetal bovine serum (FBS), and 1% antibiotic/antimycotic mix, in accordance with the standard guidelines, procedures, and protocols established by the commercial vendor. AC-16 cells were *reverse* transfected with the following vectors: *pcDNA-Ikkα-HA (K44M)* expression vector (pcDNA-Ikka-HA (K44M) was a gift from Warner Greene (Addgene plasmid # 23297), http://n2t.net/addgene:23297, 20 November 2019, RRID: Addgene_23297) [87]; *pcDNA-Ikkβ-FLAG (K44A)* expression vector (pcDNA-Ikkb-FLAG (K44A) was a gift from Warner Greene (Addgene plasmid # 23299), http://n2t.net/addgene:23299, 20 November 2019, RRID: Addgene_23299) [87]; *pCMV4-3 HA/IkB-alpha (SS32,36AA)* expression vector (pCMV4-3 HA/IkB-alpha (SS32,36AA) was a gift from Warner Greene (Addgene plasmid #24143), http://n2t.net/addgene:24143, 20 November 2019, RRID: Addgene_24143) [88]; or the corresponding pcDNA3.1 empty vector and pCMV4-3 empty vector. The transfection was performed using Polyfect^®^ (Qiagen Norge, Oslo, Norway, Catalogue # 301107) in accordance with the manufacturer’s guidelines and standardized procedures. The plasmid load to be transfected was standardized to 1 μg per 1.2 × 10^6^ cells and scaled-up or scaled-down in accordance with the stipulations of the experimental paradigm. The hypoxia–reoxygenation (H-R) challenge was effectuated by subjecting the transfected AC-16 cells to hypoxia (1% O_2_, 5% CO_2_, and 94% N_2_ for 18 h) and incubation with specific *hypoxia medium* (Appendix A) for 18 h, followed by incubation under normoxia conditions (reoxygenation) for 8 h with the standard maintenance medium. The H-R experimental paradigm is depicted in Table 1. Hypoxia was induced and maintained for the designated duration using the *New Brunswick™ Galaxy^®^ 48 R CO_2_ incubator* (Eppendorf Norge AS, Oslo, Norway). 

### 4.2. Cellular Fractionation to Segregate the Cytosolic and Nuclear Compartments

The cytosolic and nuclear fractions were isolated using the “Subcellular Protein Fractionation Kit for Cultured Cells^TM^” from Thermo Fisher Scientific (Thermo Fisher Scientific, Oslo, Norway, Catalogue # 78840) following the manufacturer’s protocol and guidelines. Briefly, AC-16 cells terminally sub-cultured and plated in 150 mm cell culture plates to the desired confluence (20 × 10^6^ cells/plate) and subjected to the respective transfection and experimental interventions (as enunciated earlier) were trypsinized and pelleted using centrifugation (1000× *g* for 5 min at 4 °C). The pelleted cells were resuspended in ice-cold 1× cytoplasmic extraction buffer (CEB) (supplied with the kit), containing DTT (dithiothreitol) and protease inhibitors, followed by incubation on ice for 15 min. The resulting cell homogenate (lysate) was transferred to a 1.5-mL microcentrifuge tube and centrifuged at 1000× *g* for 10 min at 4 °C. The resulting supernatant constituted the cytosolic fraction, and the resultant pellet was resuspended in ice-cold 1× membrane extraction buffer (MEB) (supplied with the kit), containing DTT (dithiothreitol) and protease inhibitors, followed by incubation on ice for 15 min. The resulting cell suspension was centrifuged at 3000× *g* for 5 min at 4 °C. The resulting supernatant constituted the membrane fraction, and the resultant pellet was resuspended in ice-cold 1x nuclear extraction buffer (NEB) (supplied with the kit), containing DTT (dithiothreitol) and protease inhibitors, followed by incubation on ice for 30 min and subsequent centrifugation at 5000× *g* for 5 min at 4 °C. The resulting supernatant constituted the soluble nuclear fraction, while the pellet constituted the chromatin-bound nuclear extract. The pellet was resuspended in 100 μL of room temperature 1× NEB containing 5 mM calcium chloride and 300 units of micrococcal nuclease (supplied with the kit), followed by incubation at 37 °C for 10 min and subsequent centrifugation at 16,000× *g* for 5 min at 4 °C. The resultant supernatant constituted the chromatin-bound nuclear extract.

### 4.3. Quantitative Measurement of p65 NF-κB, p50 NF-κB, HSP90-β, and histone H3 in the Cytosolic Fraction and the Nuclear Fraction Using Sandwich ELISA

The levels of p65 NF-κB and p50 NF-κB as well as HSP90β (cytosolic fraction marker) and histone H3 (nuclear fraction marker) in the cytosolic and nuclear fractions were determined by sandwich ELISA immunoassay. Briefly, 10–30 ng of the respective capture antibodies (Table 2) were immobilized in each well of the respective 96-well microplates [89]. The nuclear fractions (equivalent to 10 μg of protein content) and cytosolic fractions (equivalent to 30 μg of protein content) were incubated with the respective immobilized capture antibodies (Table 2) overnight at 4 °C. The conditioned nuclear fractions and cytosolic fractions were discarded, and the respective 96-well microplate wells were washed 3× (15 min each) with TBS-T and incubated with the respective detection antibodies (Table 2) overnight at 4 °C. The 96-well microplate wells were washed 3× (15 min each) with TBS-T followed by immunodetection with the HRP-conjugated secondary antibody, using the HRP substrate OPD (o-phenylenediamine dihydrochloride) (Thermo Fisher Scientific, Oslo, Norway, Catalogue # 34005) as a chromophore for the colorimetric read-out (λ_450_). The antibody signal specificity was established by performing *peptide-blocking assays* in the entire gamut of nuclear fractions and cytosolic fractions from experimental cells. The *antibody-blocking peptides* corresponding to the specific epitopes for the respective detection antibodies used are enumerated in Table 2. The respective absorbances from the *peptide-blocking assays* were used for the experimental blank correction. The experimental blank-corrected values were subsequently normalized and expressed as *fold change* relative to the experimental control. Data are expressed as *fold change* ± standard deviation (S.D.) from three technical replicates for each of the four biological replicates belonging to each experimental group (*n* = 4).

### 4.4. NF-κB Transcriptional Activity Reporter Assay

The transcriptional activity of NF-κB was determined in the entire gamut of experimental groups using the “NF-κB Secreted Alkaline Phosphatase (SEAPorter™) Assay Kit” from Novus Biologicals (Novus Biologicals/Bio-Techne Ltd., Abingdon, UK, Catalogue # NBP2-25286), following the manufacturer’s protocol and guidelines. Briefly, the NF-κB-SEAP expression vector was co-transfected concomitant with the other respective expression vectors (Section 2.1) in AC-16 cells, as enunciated in Section 2.1. The plasmid load of the NF-κB-SEAP expression vector was pinned to 9.6 μg per 1.2 × 10^6^ cells in adherence to the manufacturer’s protocol and guidelines and further scaled-up or scaled-down in accordance with the stipulations of the experimental paradigm. The levels of SEAP (secreted alkaline phosphatase) in the conditioned media were determined as a direct measure of NF-κB-driven expression of SEAP and as a surrogate measure of NF-κB transcriptional activity. The levels of SEAP (secreted alkaline phosphatase) in the conditioned media were determined using the SEAP substrate PNPP as the chromophore for the colorimetric read-out (λ_405_). The raw optical density values measured at λ_405_ (405 nm) were corrected with the experimental blank and subsequently normalized and expressed as *fold change* relative to the experimental control. Data are expressed as *fold change* ± standard deviation (S.D.) from three technical replicates for each of the four biological replicates belonging to each experimental group (*n* = 4).

### 4.5. Western Blotting

Proteins (10–50 μg) were resolved on SDS-PAGE gels, followed by transfer to a polyvinylidene difluoride (PVDF) membrane (Immun-Blot^TM^ PVDF Membrane, Bio-Rad Norway AS, Oslo, Norway, Catalogue # 1620177) and overnight incubation with the respective primary antibodies at 4 °C following standardized protocols [90]. The origin, source, and dilutions of the respective antibodies used in this study are compiled in Table 2. The blots were developed with enhanced chemiluminescence substrate (SuperSignal™ West Pico PLUS Chemiluminescent Substrate, Thermo Fisher Scientific, Oslo, Norway, Catalogue # 34580) and imaged using a LI-COR Odyssey XF imaging system (LI-COR Biotechnology, Cambridge, UK). Quantification of results was performed by densitometry using Image J (ImageJ, United States National Institutes of Health, Bethesda, MD, USA, https://imagej.nih.gov/ij/, 20 November 2019) [91], and the results were analyzed as total integrated densitometric values.

### 4.6. miR-210 Promoter Pull-Down Assay Using Reverse Chromatin Immunoprecipitation (R-ChIP) Approach

The miR-210 gene promoter fragment spanning 500 base pairs (bp) upstream of the TSS (transcription start site) [6,25] was affinity-captured and *pulled down* [28,29,30,92,93] to determine the relative abundance of miR-210 promoter-associated transcription factors and transcription coregulators. Total chromatin was first extracted using the “SimpleChIP^®^ Enzymatic Chromatin IP Kit” from Cell Signaling Technology (Cell Signaling Technology, Danvers, MA, USA, Catalogue # 9003) following manufacturer’s protocol and guidelines. Briefly, AC-16 cells terminally sub-cultured and plated in 150 mm cell culture plates to the desired confluence (16 × 10^6^ cells/plate) and subjected to the respective transfection and experimental interventions (as enunciated earlier) were fixed in situ with 1% formaldehyde to crosslink the chromatin-associated proteins with the chromatin. For the isolation and extraction of the crosslinked chromatin, AC-16 cells from three 150 mm plates (equitable to 48 × 10^6^ cells) were pooled together to constitute a single biological replicate within an experimental group. The isolation and extraction of the crosslinked chromatin were performed by lysing the cells and fragmenting the chromatin through partial digestion with micrococcal nuclease to achieve ~500 bp fragments. The crosslinked miR-210 promoter fragment present in the purified crosslinked chromatin was siphoned out using affinity capture on streptavidin beads by hybridization with biotin-labeled miR-210 promoter *capture probe* (Eurofins Genomics, Ebersberg, Germany). The experimental approach and strategy used to design the biotin-labeled miR-210 promoter *capture probe* are depicted in Appendix A, and the specific oligonucleotide sequence of the *capture probe* is depicted in Appendix A. The affinity-captured crosslinked miR-210 gene promoter fragment was eluted from the streptavidin beads (10 mM Tris, pH 7.5 at 90 °C for 10 min) and reverse crosslinked [94,95] in the absence of proteinase K to separate the native miR-210 gene promoter fraction from the promoter-associated proteins. The miR-210 promoter was affinity-purified using DNA spin columns provided with the kit, while the promoter-associated protein fraction was processed for Western blotting analysis, as enunciated in Section 4.5. The relative abundance of the native miR-210 gene promoter fragment was determined using tandem ELOHA (enzyme-linked oligonucleotide hybridization assay) approach [31], as described in Section 4.7.

### 4.7. Quantitative Determination of miR-210 Promoter Pull-Down Fragment Using ELOHA (Enzyme-Linked Oligonucleotide Hybridization Assay) Approach

The quantitative abundance of the miR-210 promoter *pull-down* fragment in the experimental groups was determined by adopting a novel ELOHA approach [96,97,98,99,100,101,102,103,104]. Briefly, the purified *pulled-down* miR-210 promoter (Section 4.6) was *affinity captured* on streptavidin beads using hybridization with biotin-labeled miR-210 promoter *capture probe* (Eurofins Genomics, Ebersberg, Germany). The *affinity-captured*, *pulled-down* miR-210 promoter fragment was eluted from the streptavidin beads (10 mM Tris, pH 7.5 at 90 °C for 10 min) and unsequestered from the double-stranded hybrid using *denaturation* and subsequently immobilized in the microwells of a solid phase 96-well nucleic acid microplate (Nunc™ NucleoLink™ Strips, Thermo Fisher Scientific, Oslo, Norway, Catalogue # 248259). The immobilized miR-210 was quantitated by adopting an *indirect* ELISA approach [105], whereby the immobilized miR-210 was hybridized with digoxigenin-labeled miR-210 promoter *detection probe* (Eurofins Genomics, Ebersberg, Germany), followed by immunodetection with the AP (alkaline phosphatase)-conjugated digoxigenin antibody (*Digoxigenin AP-conjugated Antibody,* R&D Systems, Minneapolis, MN, USA, Catalogue # APM7520) using the AP substrate PNPP (p-nitrophenyl phosphate, disodium salt) (Thermo Fisher Scientific, Oslo, Norway, Catalogue # 37621) as the chromophore for the colorimetric read-out (λ_405_). The experimental approach and strategy used to design the biotin-labeled miR-210 promoter *capture probe* and the *detection probe* are depicted in Appendix A. The specific oligonucleotide sequences of the respective *capture probe* and the *detection probe* are depicted in Appendix A. Competition assays were also performed with the *unlabeled detection probe* to exhibit *assay specificity* and serve as an *experimental blank*. The raw optical density values measured at λ_405_ (405 nm) were corrected with the *experimental blank* and subsequently normalized and expressed as *fold change* relative to the experimental control. Data are expressed as *fold change* ± standard deviation (S.D.) from three technical replicates for each of the four biological replicates belonging to each experimental group (*n* = 4). The strategy and approach adopted for the design of *capture* and *detection* probes for the miR-210 promoter ELOHA analysis are depicted in Appendix A.

### 4.8. Enzyme-Coupled miR-210 Hybridization Immunoassay

The levels of miR-210 in the experimental cell lysates were determined by adopting a novel microRNA immunoassay approach [96,97,98,99,100,101,102,103,104], as previously described [16,104]. This miR-210 immunoassay approach allowed the direct quantitative determination of miR-210 in the same experimental lysates being subjected to specific downstream assays. Briefly, miR-210 in the experimental lysates was *affinity captured* on streptavidin beads using hybridization with biotin-labeled miR-210 *locked nucleic acid (LNA) capture probe* (Qiagen Norge, Oslo, Norway, Catalogue # 339412 YCO0212944). The *affinity-captured* miR-210 was eluted from the streptavidin beads (10 mM Tris, pH 7.5 at 90 °C for 10 min) and unsequestered from the *double-stranded* hybrid using *denaturation* and subsequently immobilized in the microwells of a *solid phase 96-well nucleic acid microplate* (Nunc™ NucleoLink™ Strips, Thermo Fisher Scientific, Oslo, Norway, Catalogue # 248259). The immobilized miR-210 was quantitated by adopting an *indirect ELISA* approach [105], whereby the immobilized miR-210 was hybridized with digoxigenin-labeled miR-210 *LNA detection probe* (Qiagen Norge, Oslo, Norway, Catalogue # 339412 YCO0212945), followed by immunodetection with the AP (alkaline phosphatase)-conjugated digoxigenin antibody (*Digoxigenin AP-conjugated Antibody*, R&D Systems, Minneapolis, MN, USA, Catalogue # APM7520) using the AP substrate PNPP (p-nitrophenyl phosphate, disodium salt) (Thermo Fisher Scientific, Oslo, Norway, Catalogue # 37621) as the chromophore for the colorimetric read-out (λ_405_). Competition assays were also performed with the *unlabeled detection probe* to exhibit *assay specificity* and serve as an *experimental blank*. The raw optical density values measured at λ_05_ (405 nm) were corrected with the *experimental blank* and subsequently normalized and expressed as *fold change* relative to the experimental control. Data are expressed as *fold change* ± standard deviation (S.D.) from three technical replicates for each of the four biological replicates belonging to each experimental group (*n* = 4).

### 4.9. Chromatin Immunoprecipitation (ChIP) Analysis of p65 NF-κB Binding to the miR-210 Promoter

ChIP analysis was performed to determine the extent of p65 NF-κB binding to the κB-response elements in the miR-210 promoter using the “SimpleChIP^®^ Enzymatic Chromatin IP Kit” from Cell Signaling Technology (Cell Signaling Technology, Danvers, MA, USA, Catalogue # 9003) following manufacturer’s protocol and guidelines. Briefly, AC-16 cells terminally sub-cultured and plated in 150 mm cell culture plates to the desired confluence (16 × 10^6^ cells/plate) and subjected to the respective transfection and experimental interventions (as enunciated earlier) were fixed in situ with 1% formaldehyde to crosslink the chromatin-associated proteins with the chromatin. For the isolation of the chromatin, AC-16 cells from three 150 mm plates (equitable to 48 × 10^6^ cells) were pooled together to constitute a single biological replicate within an experimental group. Chromatin isolation was performed by lysing the cells and fragmenting the chromatin by partial digestion with micrococcal nuclease to achieve ~500 bp fragments. The ChIP was performed by incubating 5 µg of the ChIP-grade p65 NF-κB antibody (Table 2) with the crosslinked chromatin emanating from 16 × 10^6^ cells (one-third of the total chromatin pool of each sample). The crosslinked chromatin samples were incubated overnight at 4 °C with the designated respective ChIP-grade primary antibodies. The ChIP was also performed with normal rabbit IgG (5 µg of antibody with the crosslinked chromatin emanating from 16 × 10^6^ cells) to serve as negative control. The residual one-third of the crosslinked chromatin (from 16 × 10^6^ cells) was set aside as “input” for the final analysis. The crosslinked chromatin was reverse crosslinked [94,95] in the absence of proteinase K to separate the DNA component fraction from the chromatin-associated proteins. The DNA fraction constituting the miR-210 promoter fragment was affinity purified using DNA spin columns provided with the kit, while the chromatin-associated protein fraction was processed for Western blotting analysis, as enunciated in Section 4.3. The relative abundance of the p65 NF-κB-associated miR-210 promoter fragment was determined using tandem ELOHA (enzyme-linked oligonucleotide hybridization assay) approach [31], as described in Section 4.7. The magnitude of p65 NF-κB binding to the miR-210 promoter was expressed as *fold enrichment* by first correcting the p65 NF-κB-associated miR-210 promoter fragment ELOHA absorbance values to the respective *inputs*, followed by normalization to fold change values. 

### 4.10. Quantitative Measurement of miR-210 Promoter-Bound NF-κB Transcriptional Coactivators, p300 and CBP, as Well as NF-κB Transcriptional Corepressors, SMRT and NCoR1

The levels of p300, CBP, NCoR1, and SMRT in the reverse crosslinked miR-210 promoter *pull-down* lysates were determined using sandwich ELISA immunoassays coupled to the Biotin-Streptavidin-HRP detection system. Briefly, 20–30 ng of the respective capture antibodies (Table 2) were immobilized in each well of the respective 96-well microplates [89]. The reverse crosslinked miR-210 promoter *pull-down* lysates (equivalent to 1 μg of protein content) were incubated with the immobilized respective capture antibodies overnight at 4 °C. The conditioned lysates were discarded, and the respective 96-well microplate wells were washed 3× (15 min each) with TBS-T and incubated with the respective detection antibodies (Table 2) overnight at 4 °C. The 96-well microplate wells were washed 3× (15 min each) with TBS-T, followed by immunodetection with the HRP-conjugated secondary antibody, using the HRP substrate Amplex Red (10-acetyl-3,7-dihydroxyphenoxazine) (Thermo Fisher Scientific, Oslo, Norway, Catalogue # A22188) as the chromophore for the colorimetric read-out (λ_570_). The antibody signal specificity was established by performing *peptide-blocking assays* in the entire gamut of reverse crosslinked miR-210 promoter *pull-down* lysates. The *antibody-blocking peptides* corresponding to the specific epitopes for the respective detection antibodies used are enumerated in Table 2. The respective absorbances from the *peptide-blocking assays* were used for experimental blank correction. The experimental blank-corrected values were subsequently normalized and expressed as *fold change* relative to the experimental control. Data are expressed as *fold change* ± standard deviation (S.D.) from three technical replicates for each of the four biological replicates belonging to each experimental group (*n* = 4).

### 4.11. RNA Polymerase II (RNAPII) Occupancy Assay Using Chromatin Immunoprecipitation (ChIP)

The relative abundance of RNAPII occupancy in the miR-210 *proximal* promoter was determined using a ChIP approach, as described in Section 4.9. For the isolation of chromatin, AC-16 cells from four 150 mm plates (equitable to 60 × 10^6^ cells) were pooled together to constitute a single biological replicate within an experimental group. The ChIP was performed by incubating either 10 µg of the ChIP-grade RPB1-RNAPII antibody (Table 2) or the ChIP-grade p-Ser^2^/Ser^5^ CTD RPB-RNAPII antibody (Table 2) with the crosslinked chromatin emanating from 20 × 10^6^ cells (one-third of the total chromatin pool of each sample). The ChIP was also performed with normal mouse IgG or normal rabbit IgG (10 µg of antibody with the crosslinked chromatin emanating from 20 × 10^6^ cells) to serve as negative control. The residual one-third of the crosslinked chromatin (from 20 × 10^6^ cells) was set aside as “input” for the final analysis. The relative abundance of RPB1-RNAPII- and p-Ser^2^/Ser^5^ CTD RPB-RNAPII-associated miR-210 promoter fragments was determined using tandem ELOHA approach [31], as described in Section 4.7. The magnitude of RPB1-RNAPII and p-Ser^2^/Ser^5^ CTD RPB-RNAPII binding to the miR-210 promoter was expressed as *fold enrichment* by first correcting the RPB1-RNAPII- and p-Ser^2^/Ser^5^ CTD RPB-RNAPII-associated miR-210 promoter fragment ELOHA absorbance values to the respective *inputs*, followed by normalization to fold change values. 

### 4.12. Tandem Immunoprecipitation and Western Blotting Based Quantitative Determination of p-Thr SPT5 Expression Levels in the Reverse Crosslinked miR-210 Promoter Pull-Down Fragment

The abundance of p-Thr SPT5 in the reverse crosslinked miR-210 promoter pull-down fragment was determined using tandem immunoprecipitation and quantitative densitometry-coupled Western blot analysis. Briefly, SPT5 was immunoprecipitated out of the reverse crosslinked miR-210 promoter pull-down fragment and subjected to immunoblotting analysis with a generic antibody against phosphorylated threonine (p-Thr) residues to yield a surrogate measure of the abundance of p-Thr SPT5. For the immunoprecipitation procedure, one-third of the entire pool of the reverse crosslinked miR-210 promoter pull-down fragment was incubated with either 5 μg of SPT5 antibody (Table 2) or 5 μg of the corresponding control IgG antibody (Table 2) overnight at 4 °C. The residual one-third of the entire pool of the reverse crosslinked miR-210 promoter pull-down fragment was dedicated as the input. The respective immunocomplexes were captured and immobilized by the addition of protein A/G agarose beads and incubation overnight at 4 °C. The beads containing the immunocomplexes were washed 3× with the non-denaturing wash buffer (non-denaturing lysis buffer (20 mM Tris, 137 mM Nacl, 2 mM EDTA, 1% Nonidet P-40, 10% glycerol; pH 7.4), followed by centrifugation and discarding of the supernatant. The beads were suspended in denaturing RIPA buffer (50 mM Tris, 150 mM Nacl, 0.1% SDS, 0.5% sodium deoxycholate, 1% Triton X; pH 7.4) supplemented with protease and phosphatase inhibitors and subsequently centrifuged to pellet the beads. The supernatant containing the immunoprecipitated SPT5 was subjected to tandem Western blot analysis with the generic antibody against p-Thr residues (Table 2).

### 4.13. Histone Modification Analysis Using Chromatin Immunoprecipitation (ChIP)

Histone modification analysis delving into the acetylation and methylation status of the bona fide lysine residues in histone H3 and histone H4 was performed using a ChIP approach [106,107,108] using the “SimpleChIP^®^ Enzymatic Chromatin IP Kit” from Cell Signaling Technology (Cell Signaling Technology, Danvers, MA, USA, Catalogue # 9003) following manufacturer’s protocol and guidelines. Briefly, AC-16 cells terminally sub-cultured and plated in 150 mm cell culture plates to the desired confluence (16 × 10^6^ cells/plate) and subjected to the respective transfection and experimental interventions (as enunciated earlier) were fixed in situ with 1% formaldehyde to crosslink the chromatin-associated proteins with chromatin. For the isolation of chromatin, AC-16 cells from six 150 mm plates (equitable to 96 × 10^6^ cells) were pooled together to constitute a single biological replicate within an experimental group. Chromatin isolation was performed by lysing the cells and fragmenting the chromatin by partial digestion with micrococcal nuclease to achieve ~500 bp fragments of the chromatin. The ChIP was performed with each of the eleven designated ChIP-grade primary antibodies directed towards the following respective histone-modified residues—H3K9ac, H3K14ac, H3K18ac, H3K27ac, H4K5ac, H4K8ac, H4K12ac, H3K4me3, H3K9me3, H3K27me3, and H3K36me3—as enumerated and defined in Table 2. The ChIP was performed by incubating 5 µg of the designated respective ChIP-grade antibody with the crosslinked chromatin emanating from 6 × 10^6^ cells. The crosslinked chromatin samples were incubated overnight at 4 °C with the designated respective ChIP-grade primary antibodies. The ChIP was also performed with normal rabbit IgG (5 µg of antibody with the crosslinked chromatin emanating from 6 × 10^6^ cells) to serve as negative control. The residual one-fourth of the crosslinked chromatin (from 24 × 10^6^ cells) was set aside as “input” for the final analysis. The crosslinked chromatin was reverse crosslinked [94,95] in the absence of proteinase K to separate the DNA component fraction from the chromatin-associated proteins. The DNA fraction constituting the miR-210 promoter fragment was affinity purified using DNA spin columns provided with the kit, while the chromatin-associated protein fraction was processed for Western blotting analysis, as enunciated in Section 4.3. The relative abundance of the respective histone modification-associated miR-210 promoter fragment was determined using tandem ELOHA approach [31], as described in Section 4.7. The magnitude of the respective histone modification changes in the miR-210 promoter was expressed as *fold enrichment* by first correcting the respective histone modification-associated miR-210 promoter fragment ELOHA absorbance values to the respective *inputs*, followed by normalization to fold change values. 

### 4.14. Quantitative Measurement of HA-tag da-IκBα and HA-tag dn-IKKα/β Mutants as Well as β-Actin in Native Lysates and Loading Inputs

The levels of HA-tag *da*-IκBα and HA-tag *dn*-IKKα/β mutants, as well as β-actin in the respective native lysates and loading inputs, were determined by sandwich ELISA immunoassay. Briefly, 10–30 ng of the respective capture antibodies (Table 2) were immobilized in each well of the respective 96-well microplates [89]. The respective native lysates or the cellular *input* fractions (equivalent to 10–30 μg of protein content) were incubated with the respective immobilized capture antibodies (against the HA-tag or β-actin) (Table 2) overnight at 4 °C. The conditioned native lysates or the cellular *input* fractions were discarded, and the respective 96-well microplate wells were washed 3× (15 min each) with TBS-T and incubated with the respective detection antibodies (against the HA-tag or β-actin) (Table 2) overnight at 4 °C. The 96-well microplate wells were washed 3× (15 min each) with TBS-T, followed by immunodetection with the HRP-conjugated secondary antibody, using the HRP substrate TMB (3,3′,5,5′-tetramethylbenzidine) (Thermo Fisher Scientific, Oslo, Norway, Catalogue # N301) as a chromophore for the colorimetric read-out (λ_450_). The antibody signal specificity was established by performing *peptide-blocking assays* in the entire gamut of nuclear fractions and cytosolic fractions from experimental cells. The *antibody-blocking peptides* corresponding to the specific epitopes for the respective detection antibodies used are enumerated in Table 2. The respective absorbances from the *peptide-blocking assays* were used for experimental blank correction. The experimental blank-corrected absorbances reported were raw un-normalized (for HA-tag ELISA) or subsequently normalized and expressed as *fold change* relative to the experimental control (for β-actin ELISA). Data are expressed, either as raw un-normalized absorbance *mean* ± standard deviation (S.D.) (for HA-tag ELISA) or as *fold change* ± standard deviation (S.D.) (for β-actin ELISA) from three technical replicates for each of the four biological replicates belonging to each experimental group (*n* = 4).

### 4.15. Statistical Analysis

The significance of differences among the samples was determined by *one-way analysis of variance* (one-way ANOVA) followed by Tukey’s post hoc test. Statistical analysis was performed using GraphPad Prism 8 (GraphPad Software, San Diego, CA, USA). Quantitative data for all the assays are presented as mean values ± S.D. (mean values ± standard deviation).

## 5. Conclusions

The present study is unprecedented in characterizing the miR-210 gene promoter, as well as elucidating and delineating the molecular profile of the transcriptional regulators of miR-210 expression under cellular reoxygenation conditions. The findings and observations reported in this study unveil and delineate that the ubiquitously expressed transcription factor, NF-κB, indispensably mediates miR-210 expression during the reoxygenation phase of H-R insult. RNAPII occupancy assays and histone modification assays further unveiled the cellular constituents of the NF-κB-associated co-regulatory mechanisms and downstream molecular effectors of the H-R-driven miR-210 expression during cellular reoxygenation. This study is the first to demonstrate a direct causal relationship between H-R stress-induced NF-κB activation and H-R-evoked detrimental miR-210 expression. Furthermore, the present study is also unprecedented in characterizing miR-210 as a direct NF-κB target gene and unveiling a unique facet of the molecular underpinnings of H-R-induced miR-210 expression.

## Figures and Tables

**Figure 1 ijms-24-06618-f001:**
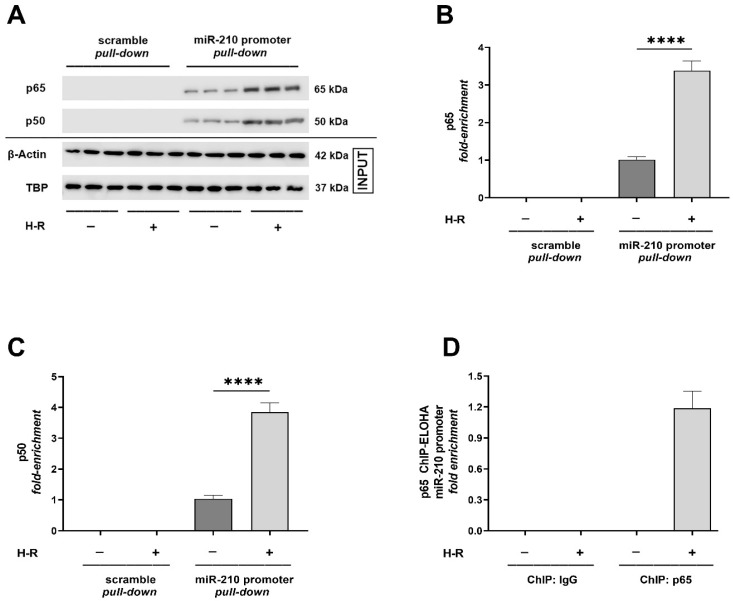
Hypoxia–reoxygenation (H-R) challenge elicits a significant increase in the association and binding of NF-κB at the miR-210 proximal promoter. (**A**–**C**) miR-210 promoter pull-down assay determining the quantitative enrichment of the p65 NF-κB and p50 NF-κB subunits in the reverse crosslinked miR-210 promoter pull-down fragment. Representative Western blots (**A**) and quantitative densitometric analysis (**B**,**C**) showing the relative quantitative abundance of p65 NF-κB and p50 NF-κB subunits in the miR-210 promoter pull-down lysates. (**D**) Tandem ChIP-ELOHA analysis showing the relative enrichment of p65 NF-κB at the κB response element in the miR-210 proximal promoter. Data from the Western blot-coupled densitometric analysis are expressed as mean *fold change* ± S.D. from three biological replicates belonging to each experimental group (*n* = 3). Data from the ChIP-ELOHA analysis are expressed as *fold enrichment* by first correcting the p65 NF-κB-associated miR-210 promoter fragment ELOHA absorbance values to the respective *inputs*, followed by normalization to fold change values. Data from the ChIP-ELOHA analysis are depicted as mean *fold enrichment* ± S.D. from three technical replicates for each of the four biological replicates belonging to each experimental group (*n* = 4). **** *p* ≤ 0.0001. S.D.: standard deviation.

**Figure 2 ijms-24-06618-f002:**
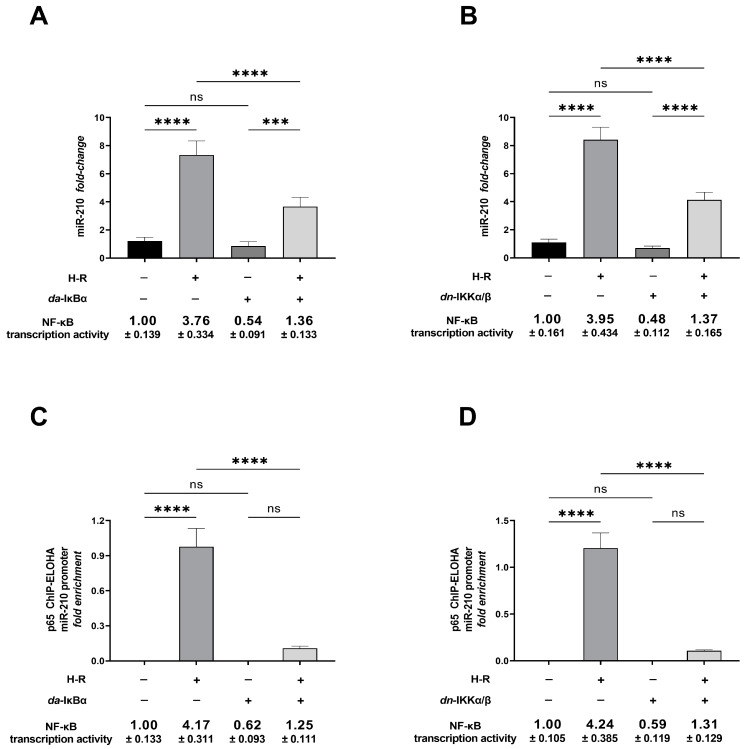
The activation and recruitment of NF-κB to the miR-210 proximal promoter is indispensable for the hypoxia–reoxygenation (H-R)-induced miR-210 expression. (**A**,**B**) Enzyme-coupled miR-210 hybridization immunoassay determining miR-210 levels in H-R challenge-subjected cells ectopically expressing either the *da*-IκBα (**A**) or *dn*-IKKα/β (**B**) mutants to inhibit NF-κB activation. (**C**,**D**) Tandem ChIP-ELOHA analysis showing the relative enrichment of p65 NF-κB at the κB response element in the miR-210 proximal promoter in response to H-R challenge-subjected cells ectopically expressing either the *da*-IκBα (**C**) or *dn*-IKKα/β (**D**) mutants to inhibit NF-κB activation. Data from miR-210 hybridization immunoassay are expressed as mean *fold change* ± S.D. from three technical replicates for each of the four biological replicates belonging to each experimental group (*n* = 4). The ectopic expression of the HA-tagged *da*-IκBα and HA-tagged *dn*-IKKα/β mutants was validated using ELISA immunoassay performed against the HA tag (Appendix A). NF-κB transcriptional activity ((**A**–**D**), bottom panel) was determined in the native lysates to corroborate and validate the translative effects of the ectopic expression of the *da*-IκBα and *dn*-IKKα/β mutants. Data from the ChIP-ELOHA analysis are expressed as *fold enrichment* by first correcting the p65 NF-κB-associated miR-210 promoter fragment ELOHA absorbance values to the respective inputs, followed by normalization to fold change values. Data from the ChIP-ELOHA analysis are depicted as mean *fold enrichment* ± S.D. from three technical replicates for each of the four biological replicates belonging to each experimental group (*n* = 4). *** *p* ≤ 0.001; **** *p* ≤ 0.0001; ns: not significant (*p* > 0.05) S.D.: standard deviation.

**Figure 3 ijms-24-06618-f003:**
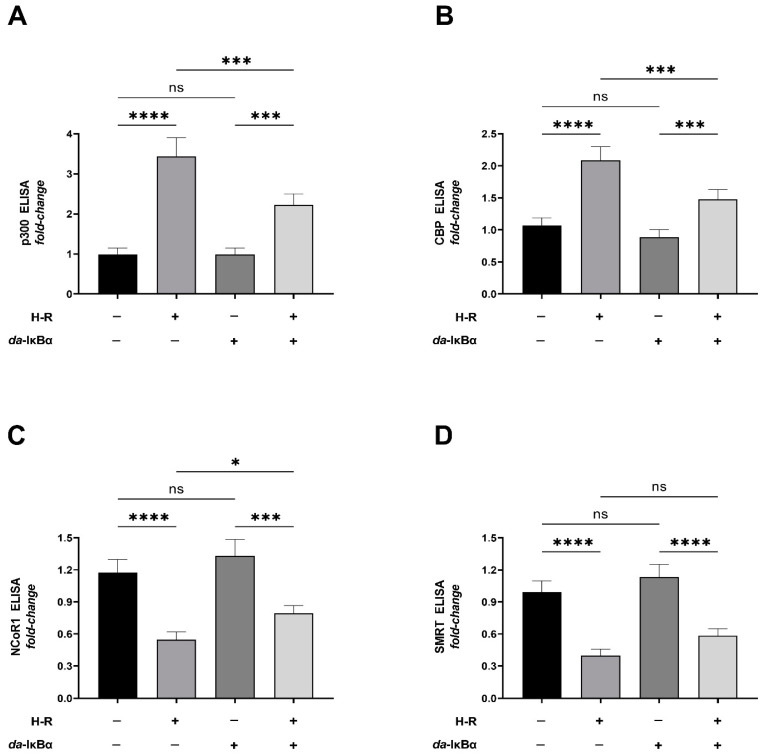
Hypoxia–reoxygenation (H-R) challenge increases the abundance of miR-210 promoter-bound transcriptional coactivators, p300 and CBP, concomitant with a decrease in abundance of the miR-210 promoter-bound transcriptional corepressors, NCoR1 and SMRT, through NF-κB activation. (**A**–**D**) Quantitative ELISA determining the relative abundance of p300 (**A**), CBP (**B**), NCoR1 (**C**), and SMRT (**D**) in the lysates emanating from the reverse crosslinked miR-210 promoter pull-down fragments from H-R challenge-subjected cells ectopically expressing the *da*-IκBα mutant. The relative abundance of β-actin (Appendix A) and chromatin-associated TBP (Appendix A) in the native cellular lysates was determined using sandwich ELISA immunoassays to validate equitable inputs across the experimental lysates subjected to miR-210 promoter pull-down. The relative abundance of chromatin-associated histone H3 (Appendix A) was determined in the reverse crosslinked miR-210 promoter pull-down fragment to validate that equitable fraction of the miR-210 promoter was pull-down across the experimental lysates. The ectopic expression of the HA-tagged *da*-IκBα mutant was validated using sandwich ELISA immunoassay performed against the HA tag (Appendix A). NF-κB transcriptional activity (Appendix A, bottom panel) was determined in the native lysates to corroborate and validate the translative effects of the ectopic expression of the *da*-IκBα mutant. Data are expressed as experimental blank-corrected absorbances (O.D) measured at λ_450_ (450 nm) normalized to *fold change* values. All data are expressed as mean *fold change* ± S.D. values from three technical replicates for each of the four biological replicates belonging to each experimental group (*n* = 4). * *p* ≤ 0.05; *** *p* ≤ 0.001; **** *p* ≤ 0.0001; ns: not significant (*p* > 0.05). S.D.: standard deviation.

**Figure 4 ijms-24-06618-f004:**
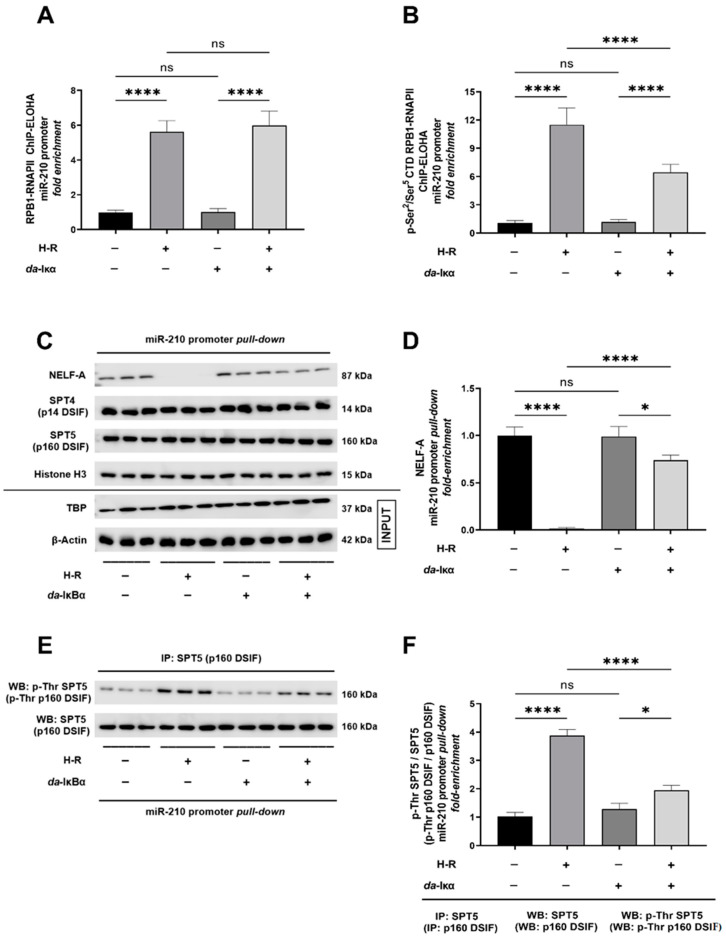
NF-κB activation significantly mediates the hypoxia–reoxygenation (H-R) challenge-induced increase in the recruitment and occupancy of *active* RNA polymerase II (RNAPII) at the miR-210 promoter. (**A**,**B**) Tandem ChIP-ELOHA analysis showing the relative enrichment of the largest subunit and the catalytic component of RNAPII, RPB1 (**A**), and the CTD-phosphorylated Ser^2^/Ser^5^ RPB1 (**B**) in the miR-210 proximal promoter in response to H-R challenge-subjected cells ectopically expressing the *da*-IκBα mutant. Data from the ChIP-ELOHA analysis are expressed as *fold enrichment* by first correcting the RPB1-associated miR-210 promoter fragment ELOHA absorbance values to the respective *inputs*, followed by normalization to fold change values. The ectopic expression of the HA-tagged *da*-IκBα mutant was validated using sandwich ELISA immunoassay performed against the HA tag (Appendix A). NF-κB transcriptional activity (Appendix A) was determined in the native lysates to corroborate and validate the translative effects of the ectopic expression of the *da*-IκBα mutant. Data from the ChIP-ELOHA analysis are depicted as mean *fold enrichment* ± S.D. from three technical replicates for each of the four biological replicates belonging to each experimental group (*n* = 4). (**C**,**D**) Representative Western blots (**C**) and quantitative densitometric analysis (**D**) showing the relative quantitative abundance of the molecular components of the NELF and DSIF protein complexes associated with the RPB1 subunit recruited at the miR-210 proximal promoter in response to H-R challenge-subjected cells ectopically expressing the *da*-IκBα mutant. (**E**,**F**) Representative Western blots (**E**) and quantitative densitometric analysis (**F**) of the abundance of p-Thr residues in the immunoprecipitated SPT5 as a surrogate measure of the status of RNAPII-mediated transcription elongation in response to H-R challenge-subjected cells ectopically expressing the *da*-IκBα mutant. Data from the Western blot and densitometric analysis are expressed as mean *fold change* ± S.D. from three biological replicates belonging to each experimental group (*n* = 3). * *p* ≤ 0.05; **** *p* ≤ 0.0001; ns: not significant (*p* > 0.05). S.D.: standard deviation.

**Figure 5 ijms-24-06618-f005:**
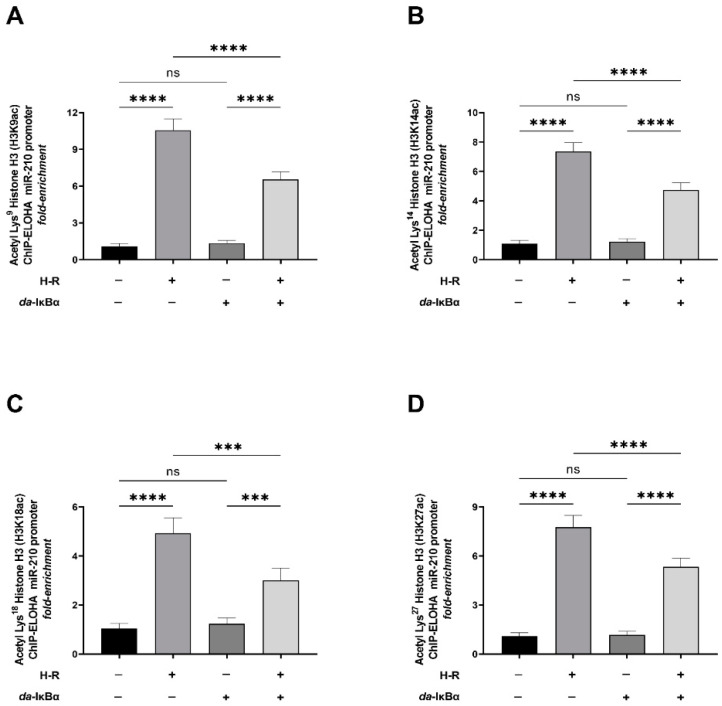
NF-κB activation significantly mediates the hypoxia–reoxygenation (H-R) challenge-induced increase in acetylation of characteristic lysine residues in histone H3 that are signatory of an open *permissive* and transcriptionally active miR-210 promoter. (**A**–**D**) Tandem ChIP-ELOHA analysis showing the relative abundance of histone H3 acetylated at (**A**) Lys^9^—H3K9ac; (**B**) Lys^14^—H3K14ac; (**C**) Lys^18^—H3K18ac; and (**D**) Lys^27^—H3K27ac in the miR-210 proximal promoter in response to H-R challenge-subjected cells ectopically expressing the *da*-IκBα mutant. Data from the ChIP-ELOHA analysis are expressed as *fold enrichment* by first correcting the respective acetylated histone H3-associated miR-210 promoter fragment ELOHA absorbance values to the respective *inputs*, followed by normalization to fold change values. Data from the ChIP-ELOHA analysis are depicted as mean *fold enrichment* ± S.D. from three technical replicates for each of the four biological replicates belonging to each experimental group (*n* = 4). *** *p* ≤ 0.001; **** *p* ≤ 0.0001; ns: not significant (*p* > 0.05). S.D.: standard deviation.

**Figure 6 ijms-24-06618-f006:**
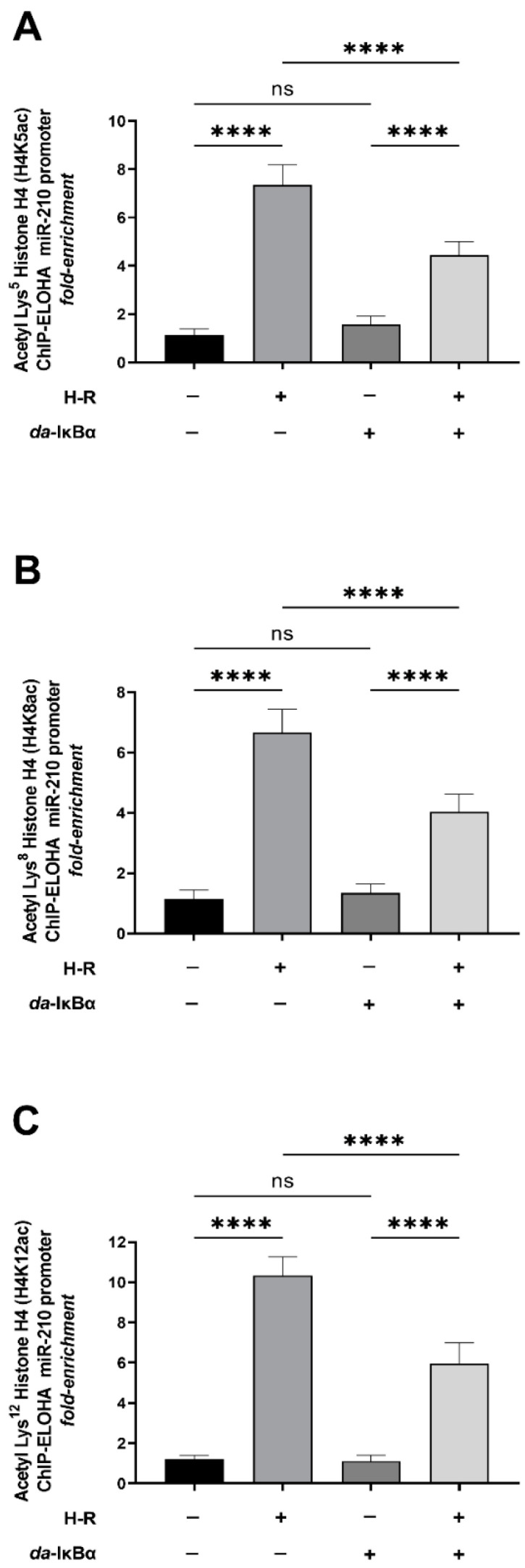
Hypoxia–reoxygenation (H-R) challenge-induced increase in acetylation of characteristic lysine residues in histone H4 that are signatory of an open *permissive* and transcriptionally active miR-210 promoter is significantly dependent on NF-κB activation. (**A**–**C**) Tandem ChIP-ELOHA analysis showing the relative abundance of histone H4 acetylated at (**A**) Lys^5^—H4K5ac; (**B**) Lys^8^—H4K8ac; and (**C**) Lys^12^—H4K12ac in the miR-210 proximal promoter in response to H-R challenge-subjected cells ectopically expressing the *da*-IκBα mutant. Data from the ChIP-ELOHA analysis are expressed as *fold enrichment* by first correcting the respective acetylated histone H4-associated miR-210 promoter fragment ELOHA absorbance values to the respective *inputs*, followed by normalization to fold change values. Data from the ChIP-ELOHA analysis are depicted as mean *fold enrichment* ± S.D. from three technical replicates for each of the four biological replicates belonging to each experimental group (*n* = 4). **** *p* ≤ 0.0001; ns: not significant (*p* > 0.05). S.D.: standard deviation.

**Figure 7 ijms-24-06618-f007:**
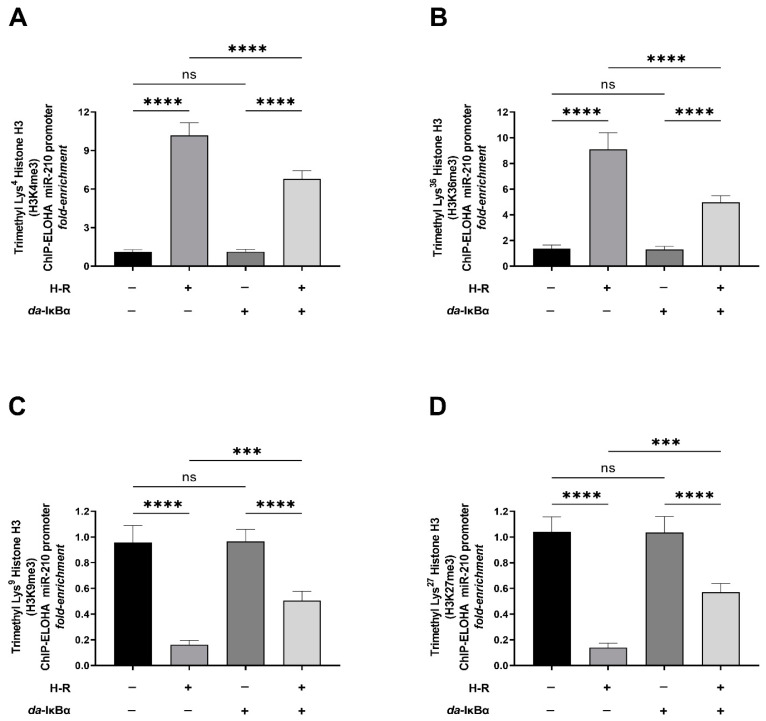
NF-κB activation significantly mediates the hypoxia–reoxygenation (H-R) challenge-induced changes in methylation of characteristic lysine residues in histone H3 that are signatory of an open *permissive* and transcriptionally active miR-210 promoter. (**A**,**B**) Tandem ChIP-ELOHA analysis of the miR-210 proximal promoter showing the relative abundance of histone H3 trimethylated at (**A**) Lys^4^—H3K4me3 and (**B**) Lys^36^—H3K36me3, a molecular hallmark of a *permissive* active promoter that is associated with transcriptional activation. (**C**,**D**) Tandem ChIP-ELOHA analysis of the miR-210 proximal promoter showing the relative abundance of histone H3 trimethylated at (**A**) Lys^9^—H3K9me3 and (**B**) Lys^27^—H3K27me3, a molecular hallmark of a *refractory* promoter that is associated with transcriptional repression. Data from the ChIP-ELOHA analysis are expressed as *fold enrichment* by first correcting the respective acetylated histone H4-associated miR-210 promoter fragment ELOHA absorbance values to the respective *inputs*, followed by normalization to fold change values. Data from the ChIP-ELOHA analysis are depicted as mean *fold enrichment* ± S.D. from three technical replicates for each of the four biological replicates belonging to each experimental group (*n* = 4). *** *p* ≤ 0.001; **** *p* ≤ 0.0001; ns: not significant (*p* > 0.05). S.D.: standard deviation.

**Table 1 ijms-24-06618-t001:** Experimental paradigm and experimental groups.

	pCMV4-3HA Empty Vector (pCMV4-3HA EV)	pCMV4-3 HA/IκB-Alpha (SS32,36AA) = Denoted *da*-IκBα	pcDNA3.1 Empty Vector (pcDNA3.1 EV)	*pcDNA-Ikkα-HA (K44M) + pcDNA-Ikkβ-FLAG (K44A)* = Denoted *dn*-IKKα/β
Normoxia, (18 h + 8 h)	*n* = 4	*n* = 4	*n* = 4	*n* = 4
H-R [Hypoxia (18 h) + Reoxygenation (8 h)]	*n* = 4	*n* = 4	*n* = 4	*n* = 4

*n* = 4: four biological replicates.

**Table 2 ijms-24-06618-t002:** List of antibodies and antibody-blocking peptides used in the study.

Antibody	Application	Amount	Host	Manufacturer	Catalogue #
β-Actin	WB 1:5000	1 µg	Mouse	Santa Cruz Biotechnology, Dallas, TX, USA	sc-47778
β-Actin	ELISA capture	20 ng/well	Mouse	Santa Cruz Biotechnology, Dallas, TX, USA	sc-47778
β-Actin	ELISA detection	20 ng/well	Rabbit	Cell Signaling Technology, Danvers, MA, USA	4970
β-Actin Antibody-Blocking Peptide	ELISA detection	N/A	N/A	Cell Signaling Technology, Danvers, MA, USA	1025
CBP	ELISA capture	30 ng/well	Mouse	Thermo Fisher Scientific, Waltham, MA, USA	H00001387-M02
CBP	ELISA detection	30 ng/well	Rabbit	Novus Biologicals, Abingdon, UK	NBP2-38774
CBP Antibody-Blocking Peptide	ELISA detection	N/A	N/A	Novus Biologicals, Abingdon, UK	NBP2-38774PEP
Goat Anti-Mouse IgG (H + L)–HRP Conjugate	1:5000	1 µg	Goat	Bio-Rad, Hercules, CA, USA	1706516
Goat Anti-Mouse IgG–AP Conjugate	1:5000	N/A ^€^	Goat	Bio-Rad, Hercules, CA, USA	1706520
Goat Anti-Rabbit IgG (H + L)–HRP Conjugate	1:5000	1 µg	Goat	Bio-Rad, Hercules, CA, USA	1706515
Goat Anti-Rabbit IgG–AP Conjugate	1:20000	N/A ^€^	Goat	Sigma Aldrich/Merck Life Science, Darmstadt, Germany	A3687
HA tag	WB 1:1000	5 µg	Mouse	Thermo Fisher Scientific, Waltham, MA, USA	26183
HA tag	ELISA capture	30 ng/well	Mouse	Thermo Fisher Scientific, Waltham, MA, USA	26183
HA tag	ELISA detection	30 ng/well	Rabbit	Abcam, Cambridge, UK	ab13834
HA-Tag Antibody-Blocking Peptide	ELISA detection	N/A	N/A	Abcam, Cambridge, UK	ab13835
HSP90-β	ELISA capture	30 ng/well	Mouse	Novus Biologicals, Abingdon, UK	NBP2-37590
HSP90-β	ELISA detection	40 ng/well	Rabbit	Novus Biologicals, Abingdon, UK	NBP2-68978
HSP90-β Antibody-Blocking Peptide	ELISA detection			Novus Biologicals, Abingdon, UK	NBP2-68978PEP
Acetyl–Lys^9^ Histone H3 (H3K9ac)	ChIP	15 µg	Rabbit	Cell Signaling Technology, Danvers, MA, USA	9649
Acetyl–Lys^14^ Histone H3 (H3K14ac)	ChIP	15 µg	Rabbit	Cell Signaling Technology, Danvers, MA, USA	7627
Acetyl–Lys^18^ Histone H3 (H3K18ac)	ChIP	15 µg	Rabbit	Cell Signaling Technology, Danvers, MA, USA	13998
Acetyl–Lys^27^ Histone H3 (H3K27ac)	ChIP	15 µg	Rabbit	Cell Signaling Technology, Danvers, MA, USA	8173
Acetyl–Lys^5^ Histone H4 (H4K5ac)	ChIP	15 µg	Rabbit	Cell Signaling Technology, Danvers, MA, USA	8647
Acetyl–Lys^8^ Histone H4 (H4K8ac)	ChIP	15 µg	Rabbit	Cell Signaling Technology, Danvers, MA, USA	2594
Acetyl–Lys^12^ Histone H4 (H4K12ac)	ChIP	15 µg	Rabbit	Cell Signaling Technology, Danvers, MA, USA	13944
trimethyl-Lys^4^ Histone H3 (H3K4me3)	ChIP	15 µg	Rabbit	Cell Signaling Technology, Danvers, MA, USA	9751
Trimethyl–Lys^9^ Histone H3 (H3K9me3)	ChIP	15 µg	Rabbit	Cell Signaling Technology, Danvers, MA, USA	13969
Trimethyl–Lys^27^ Histone H3 (H3K27me3)	ChIP	15 µg	Rabbit	Cell Signaling Technology, Danvers, MA, USA	9733
Trimethyl–Lys^36^ Histone H3 (H3K36me3)	ChIP	15 µg	Rabbit	Cell Signaling Technology, Danvers, MA, USA	4909
Mouse IgG	ChIP, IP	5–15 µg	Mouse	Santa Cruz Biotechnology, Dallas, TX, USA	sc-2025
NCoR1	ELISA capture	30 ng/well	Mouse	Thermo Fisher Scientific, Waltham, MA, USA	MA5-15447
NCoR1	ELISA detection	30 ng/well	Rabbit	Thermo Fisher Scientific, Waltham, MA, USA	PA1-844A
NCoR1 Antibody-Blocking Peptide	ELISA detection	N/A	N/A	Thermo Fisher Scientific, Waltham, MA, USA	PEP-061
NELF-A (WHSC2)	WB	5 µg	Mouse	Thermo Fisher Scientific, Waltham, MA, USA	MA5-17199
p50 NF-κB	ELISA capture	30 ng/well	Mouse	Cell Signaling Technology, Danvers, MA, USA	13681
p50 NF-κB	ELISA detection	20 ng/well	Rabbit	Novus Biologicals, Abingdon, UK	NBP1-87758
p50 NF-κB Antibody-Blocking Peptide	ELISA detection	N/A	N/A	Novus Biologicals, Abingdon, UK	NBP1-87758PEP
p65 NF-κB	ChIP	15 µg	Mouse	Cell Signaling Technology, Danvers, MA, USA	6956
p65 NF-κB	ELISA capture	30 ng/well	Mouse	Cell Signaling Technology, Danvers, MA, USA	6956
p65 NF-κB	ELISA detection	20 ng/well	Rabbit	Novus Biologicals, Abingdon, UK	NBP2-24541
p65 NF-κB Antibody-Blocking Peptide	ELISA detection	N/A	N/A	Novus Biologicals, Abingdon, UK	NBP2-24541PEP
p300	ELISA capture	20 ng/well	Mouse	Thermo Fisher Scientific, Waltham, MA, USA	H00002033-M02
p300	ELISA detection	20 ng/well	Rabbit	Novus Biologicals, Abingdon, UK	NBP1-87693
p300 Antibody-Blocking Peptide	ELISA detection	N/A	N/A	Novus Biologicals, Abingdon, UK	NBP1-87693PEP
p-Thr	WB of IP:SPT5	5 µg	Mouse	Cell Signaling Technology, Danvers, MA, USA	9386
Rabbit IgG	ChIP, IP	5–15 µg	Rabbit	Cell Signaling Technology, Danvers, MA, USA	2729
RPB1-RNAPII	ChIP	10 µg	Mouse	Cell Signaling Technology, Danvers, MA, USA	2629
p-Ser^2^/Ser^5^ CTD RPB1-RNAPII	ChIP	10 µg	Rabbit	Cell Signaling Technology, Danvers, MA, USA	13546
SMRT	ELISA capture	20 ng/well	Rabbit	Thermo Fisher Scientific, Waltham, MA, USA	BS-1420R
SMRT	ELISA detection	20 ng/well	Mouse	Thermo Fisher Scientific, Waltham, MA, USA	MA1-843
SMRT Antibody-Blocking Peptide	ELISA detection	N/A	N/A	Thermo Fisher Scientific, Waltham, MA, USA	PEP-043
SPT4	WB	5 µg	Rabbit	Thermo Fisher Scientific, Waltham, MA, USA	PA5-103308
SPT5	IP	5 µg	Rabbit	Thermo Fisher Scientific, Waltham, MA, USA	A300-868A
SPT5	WB	5 µg	Mouse	Thermo Fisher Scientific, Waltham, MA, USA	H00006829-M03
TBP	WB	5 µg	Mouse	Thermo Fisher Scientific, Waltham, MA, USA	49-1036
TBP	ELISA capture	20 ng/well	Mouse	Thermo Fisher Scientific, Waltham, MA, USA	49-1036
TBP	ELISA detection	20 ng/well	Rabbit	Novus Biologicals, Abingdon, UK	NBP2-38610
TBP Antibody-Blocking Peptide	ELISA detection	N/A	N/A	Novus Biologicals, Abingdon, UK	NBP2-38610PEP

WB: Western blot, ChIP: chromatin immunoprecipitation, IP: immunoprecipitation, N/A: not available/not applicable, €: amount of secondary antibody cannot be determined as the commercial vendor does not provide the antibody concentration.

## Data Availability

All data are included in the manuscript.

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
