# Peer review of "NF-κB Transcriptional Activity Indispensably Mediates Hypoxia–Reoxygenation Stress-Induced microRNA-210 Expression"

_ijms, 2023, doi:10.3390/ijms24076618_

Round 1

Reviewer 1 Report

Dear Authors,

Hypoxia-reoxygenation underlies many pathological conditions. In this regard, numerous attempts are being made and various studies are being carried out on the molecular mechanisms associated with this condition. However, there are still many gaps in the study of damage development pathways and molecular targets aimed at its prevention. Your research demonstrated a fundamental and well-structured approach to the studying such targets by identifying the miR-210/NF-κB regulatory axis in enhancing cardiomyocyte death under reoxygenation conditions.

This is excellent work, bringing together a whole set of evidence-based experimental approaches.

As a wish for other studies in the future. It would be interesting to explore other NF-κB-targeting hypoxamiRs in the context of studying their relationship in ischemia.

Author Response

Hypoxia-reoxygenation underlies many pathological conditions. In this regard, numerous attempts are being made and various studies are being carried out on the molecular mechanisms associated with this condition. However, there are still many gaps in the study of damage development pathways and molecular targets aimed at its prevention. Your research demonstrated a fundamental and well-structured approach to the studying such targets by identifying the miR-210/NF-κB regulatory axis in enhancing cardiomyocyte death under reoxygenation conditions.

This is excellent work, bringing together a whole set of evidence-based experimental approaches.

We sincerely thank the reviewer for critically reviewing our study and providing a constructive and positive feed-back. We sincerely thank the reviewer for appreciating our work and the positive appraisal of the findings and observations reported in our manuscript. We deeply value and cherish the reviewer’s appreciation of our work.

As a wish for other studies in the future. It would be interesting to explore other NF-κB-targeting hypoxamiRs in the context of studying their relationship in ischemia.

We completely agree with the reviewer. Indeed, given that NF-κB is a pivotal signaling molecule that regulates and modulates multiple disparate facets of the cellular biochemical response to ischemia-reperfusion injury, further studies are warranted in determining the role of NF-κB signaling in the transcriptional induction of other known hypoxamiRs (miR-23, miR-24, miR-26, miR-27, miR-103, miR-107, miR-181, and miR-213), both under hypoxia/ischemia stress and during cellular reoxygenation / tissue reperfusion scenarios. 

Reviewer 2 Report

The manuscript by Gurdeep Marwarha et al. investigates the transcriptional induction of miR-210 in AC16 cardiomyocytes exposed to H/R, establishing NF-κB as an up-stream regulator of the augmented miR-210 expression during cellular reoxygenation.

The study is original and very relevant for this scientific area. It is noteworthy that NF-kB is indispensable for miR-210 expression induced by H/R, during the reoxygenation phase.

The experimental plan is well designed. The results are presented in a clear and detailed manner.

The conclusions are strongly supported by the results.

 The manuscript is well written and it guides the reader through a complex topic.

I have only few minor comments:

While you mention in the abstract the role of NF-kB in the induction of mir-210 specifically during cellular reoxygenation, in the discussion you describe NF-κB as a significant modulator of miR-210 expression in response to cellular H-R challenge. It could be relevant highlighting in the discussion that HIF-1α mediates the expression of miR-210 during hypoxia, while NF-κB during cellular reoxygenation.

Introduction pg 2: please provide a reference for this sentence:when HIF1α expression and transcriptional activity is completely abrogated”.

Results:

Paragraph 2.1, Line 13 “validated NF-κB activation as a molecular consequence of H-R AC-.” Please check this sentence for typo (AC-).

Fig. 3D: The quantitative ELISA determining the relative abundance of SMRT shows no significant difference between H-R challenge with or without da-IkBα. Please modify the text of the results accordingly (pg. 7).

Fig. 4C and 4E: Is the label indicating the absence and the presence of da-Ikα missing? Otherwise, please explain the experiment.

Supplementary figures are indicated in the figure legends as “Scheme”. Please modify the supplementary figure legends according to the main text of the manuscript.

Author Response

The manuscript by Gurdeep Marwarha et al. investigates the transcriptional induction of miR-210 in AC16 cardiomyocytes exposed to H/R, establishing NF-κB as an up-stream regulator of the augmented miR-210 expression during cellular reoxygenation. The study is original and very relevant for this scientific area. It is noteworthy that NF-kB is indispensable for miR-210 expression induced by H/R, during the reoxygenation phase. The experimental plan is well designed. The results are presented in a clear and detailed manner. The conclusions are strongly supported by the results. The manuscript is well written and it guides the reader through a complex topic.                                                                                                                                                        We sincerely thank the reviewer for critically reviewing our study and providing a constructive and positive feed-back. We sincerely thank the reviewer for appreciating our work and the positive appraisal of the findings and observations reported in our manuscript. We deeply value and cherish the reviewer’s appreciation of our work.

I have only few minor comments:

While you mention in the abstract the role of NF-kB in the induction of mir-210 specifically during cellular reoxygenation, in the discussion you describe NF-κB as a significant modulator of miR-210 expression in response to cellular H-R challenge. It could be relevant highlighting in the discussion that HIF-1α mediates the expression of miR-210 during hypoxia, while NF-κB during cellular reoxygenation.         

We completely agree with the reviewer. We have added the following excerpt in the Discussion section that emphasizes on the distinction between the hypoxia-driven HIF1α-mediated miR-210 expression versus the cellular reoxygenation-driven NF-κB – mediated miR-210 expression. The excerpt is now embedded in the Discussion section (page 16, lines 22-29) and reads as follows,

“Ergo, it is imperative to distinguish between the hypoxia / ischemia – elicited transcriptional induction of miR-210 versus the cellular reoxygenation / tissue reperfusion – evoked transcriptional induction of miR-210. While the hypoxia / ischemia – evoked HIF1α-mediated transcriptional induction of miR-210 has been exhaustively characterized and well delineated (Ivan et al., 2001; Jiang et al., 1997; Lee et al., 2004; Wang et al., 1995), the unprecedented findings from this study characterized an implicit role of NF-κB as an indispensable molecular entity and transcriptional component that orchestrates the cellular reoxygenation – elicited transcriptional induction of miR-210.”

Also, throughout the Discussion section, where most appropriate, we have replaced “H-R insult / H-R challenge” with the phrase “cellular reoxygenation” to put a greater emphasis and confer additional clarity to the temporal context of the experimental end-point, i.e., cellular reoxygenation. 

Introduction pg 2: please provide a reference for this sentence: “when HIF1α expression and transcriptional activity is completely abrogated”.                                                                                    

We have now added four (4) references, both primary studies and leading comprehensive reviews, that establish and lend support to our contemporary understanding that HIF1α expression and transcriptional activity is nonexistent under normoxia conditions. These references are enumerated below and are inserted in the main body of the “Introduction” section. They have not been formatted in adherence to the IJMS guidelines and stipulations, such that they overtly stand-out and thus could be readily noticed by the reviewers and the editorial team. The references are as follows,

 “when HIF1α expression and transcriptional activity is completely abrogated (Ivan et al., 2001; Jiang et al., 1997; Lee et al., 2004; Wang et al., 1995).”

Ivan, M., Kondo, K., Yang, H., Kim, W., Valiando, J., Ohh, M., Salic, A., Asara, J. M., Lane, W. S., & Kaelin Jr, W. G. (2001). HIFα Targeted for VHL-Mediated Destruction by Proline Hydroxylation: Implications for O2 Sensing. Science, 292(5516), 464-468. https://doi.org/10.1126/science.1059817

Jiang, B.-H., Zheng, J. Z., Leung, S. W., Roe, R., & Semenza, G. L. (1997). Transactivation and Inhibitory Domains of Hypoxia-inducible Factor 1α: MODULATION OF TRANSCRIPTIONAL ACTIVITY BY OXYGEN TENSION*. Journal of Biological Chemistry, 272(31), 19253-19260. https://doi.org/https://doi.org/10.1074/jbc.272.31.19253

Lee, J.-W., Bae, S.-H., Jeong, J.-W., Kim, S.-H., & Kim, K.-W. (2004). Hypoxia-inducible factor (HIF-1)α: its protein stability and biological functions. Experimental & Molecular Medicine, 36(1), 1-12. https://doi.org/10.1038/emm.2004.1

Wang, G. L., Jiang, B. H., Rue, E. A., & Semenza, G. L. (1995). Hypoxia-inducible factor 1 is a basic-helix-loop-helix-PAS heterodimer regulated by cellular O2 tension. Proc Natl Acad Sci U S A, 92(12), 5510-5514. https://doi.org/10.1073/pnas.92.12.5510

Results: Paragraph 2.1, Line 13 “validated NF-κB activation as a molecular consequence of H-R AC-.” Please check this sentence for typo (AC-).                                                                                   

We thank the reviewer for highlighting this typographical error “AC-”. We have now corrected the typographical error by deleting “AC-” from the respective sentence.

Fig. 3D: The quantitative ELISA determining the relative abundance of SMRT shows no significant difference between H-R challenge with or without da-IkBα. Please modify the text of the results accordingly (pg. 7).                                                                                                              

We absolutely agree with the reviewer and thank the reviewer’s insight for identifying this inconsistency and inaccuracy in the data-description of “Figure 3D” in the “Results” section. We have now corrected the error and amended the excerpt describing “Figure 3D” in the “Results” section such that it accurately describes the data depicted in Figure 3D.

Fig. 4C and 4E: Is the label indicating the absence and the presence of da-Ikα missing? Otherwise, please explain the experiment.                                                                                                                  

We thank the reviewer’s observation and insight for identifying this error in the labeling of the western blot panels depicted in Figure 4C and Figure 4E. The experiment in Figure 4C and Figure 4E is analogous to the experiments depicted in other panels of Figure 4 (and throughout the manuscript). We have now corrected this labeling error and amended the panels, Figure 4C and Figure 4E, such that it includes the label “absence and the presence of da-Ikα” consistent with the other panels depicted in Figure 4.

Supplementary figures are indicated in the figure legends as “Scheme”. Please modify the supplementary figure legends according to the main text of the manuscript.                                          

We have now amended the Supplementary Figure legends such that the indication “Scheme” is changed to “Supplementary Figure” (for example changed Scheme 1 to Supplementary Figure 1), in all the seven (7) supplementary figures, page 28 through page 34.